# Effects of Low Benzoic Acid Concentrations on Growth and Substrate Utilization in Black Soldier Fly Larvae

**DOI:** 10.3390/insects16111155

**Published:** 2025-11-12

**Authors:** Thor Brødsted Christiansen, Niels Thomas Eriksen

**Affiliations:** Department of Chemistry and Bioscience, Aalborg University, Fredrik Bajers Vej 7H, DK-9220 Aalborg, Denmark; thorbchristiansen@gmail.com

**Keywords:** BSF larvae, *Hermetia illucens*, antimicrobial agent, substrate pH, larval density, substrate reduction rate, substrate conversion efficiency, net growth efficiency

## Abstract

Black soldier fly (BSF) larvae are reared on moist, organic materials. The larvae live mixed in with their feed substrate, where larvae and microorganisms jointly degrade substrate components and compete for nutritional compounds. Modulation of the microbial activities inside the feed substrates may thus affect the observed feed efficiencies of the BSF larvae. Here, we evaluate the effect of low concentrations of benzoic acid, a commonly used antimicrobial agent in foods, on the performance of BSF larvae reared on chicken feed. Benzoic acid at concentrations of 0–0.2% did not affect the BSF larvae negatively but lowered the microbial substrate degradation, thus leading to improved feed efficiencies. However, benzoic acid was active only when pH of the feed substrates was lowered to levels, negatively affecting the performance of the larvae. Still, this study demonstrates that antimicrobial agents can be used in BSF farming to enhance recycling of organic materials into insect biomass.

## 1. Introduction

Larvae of the black soldier fly (BSF) can convert a wide range of organic residues into food, feed, fertilizer, and other commodities and have become one of the most farmed insects in many parts of the world [1]. The feed efficiency, that is the efficiency by which the larvae convert feed substrates into larval biomass is, however, highly variable. Variability arises from differences in larval density [2,3,4], the quality and composition of the feed substrates [5], and variations in larval metabolic rates [6,7]. Also, concurrent microbial activities in the feed substrates may affect the feed efficiency of BSF larvae. BSF larvae are reared on moist, organic feed substrates, in which microorganisms play different roles. They can improve the availability of feed components by co-digestion of recalcitrant polymeric molecules, thus benefiting the BSF larvae [8,9,10]. However, also feed-grade materials and food waste, which are readily digested by BSF larvae, are often included in the feed substrates for BSF larvae [1,11]. In such feed substrates, microorganisms may compete with the BSF larvae for feed, not least if the substrate moisture content is low and O_2_ can diffuse into the feed substrates [11], or if the density of the feed substrate is low with air pockets inside [12]. In those cases, microbial metabolism, measured in terms of CO_2_ emissions, can exceed the activity of BSF larvae, reducing the availability of food components to the larvae, lowering their maximum weight, overall yield, and feed efficiency. In wetter and more compact feed substrates, microbial CO_2_ emissions are much lower [11,12]. Microbial activities are then likely dominated by anaerobic processes that produce high concentrations of lactic acid and other fermentation products [13], which may preserve the feed substrate from further microbial degradation and remain available as food for the BSF larvae. Still, minor effects from microbial activities on larval performance are to be expected, since the microorganisms, in all cases, act as additional trophic levels inside the feed substrates, inevitably adding to the metabolic loss of feed substrates.

Different performance indicators are used to quantify feed efficiencies. The substrate conversion efficiency (SCE) is the ratio between the increase in total larval weight and the weight loss of the feed substrates across the rearing process. The SCE is evaluated based on start-to-end masses of the larvae and the feed substrates and frass, and it represents the combined performance of the larvae and the microorganisms in the feed substrates. The average carbon net growth efficiency (NGE*_avg_) is a second performance indicator which depends solely on the metabolic performance of the larvae. NGE*_avg_ describes the ratio between how much carbon is stored and how much carbon has been assimilated by the larvae across the rearing process [6,14]. To better compare the different performance indicators, NGE*_avg_ can be expressed also in terms of dry matter (NGE*_avg,DW_) if the carbon contents of larvae and feed substrates are taken into consideration [7]. NGE*_avg,DW_ marks the upper limit for the SCE. Comparatively low SCEs can therefore indicate losses of feed substrates due to concurrent microbial activities. Indeed, SCE is often considerably smaller than NGE*_avg,DW_ or NGE*_avg_ and also more variable [7,14]. Sometimes, feed utilization is also reported using the substrate reduction rate (SRR), which is the ratio between the used and initial amount of feed substrate. The SRR has also been highly variable across studies on BSF larvae reared on chicken feed, with values 0.43–0.85, further indicating that the degradation of feed substrates may be affected not just by the BSF larvae [14].

Reduced microbial activities in the feed substrates can increase feed efficiencies and larval yields [12]. Microbial activities can be reduced by antimicrobial agents, which must, however, be harmless to the BSF larvae and not compromise safety for them to be relevant in BSF farming. Potassium sorbate inhibits the formation of mold in feed substrates for BSF larvae, which apparently increases larval weight and improves overall larval mass production [15]. Potassium sorbate also lowers the temperature of BSF larval feed substrates, indicating a general decrease in microbial activity in the feed substrates, but may also make BSF larvae grow slower [16]. Various other natural antimicrobial compounds and extracts are used for preservation of food, posing no health risks to human consumers [17]. One is benzoic acid (E210) or sodium benzoate (E211). This is a widely used preservative in food and feed due to its antimicrobial properties. Benzoic acid is a weak acid with a p*K_a_* value of 4.2. It inhibits the growth of yeast, mold, and some bacteria, particularly at low pH [18]. Undissociated benzoic acid is the active component that penetrates microbial cells, lowers their intracellular pH, and disrupts their metabolism [19]. Benzoic acid is also used as feed additive to livestock [20], evaluated safe to use in doses of 0.5–10 g kg^−1^ (0.05–1%) in feeds for poultry and pigs [21,22]. Positive effects from benzoic acid on gut health, gut microflora, and feed utilization have been reported in fish [23], chicken [24], and mammals [25,26]. Benzoic acid has proven less toxic to nematodes compared to other organic acids [27]. On the other hand, fruit fly larvae are delayed in their development by sodium benzoate at concentrations above 0.1% [28], and similar benzoic acid concentrations reduce the number of adult emergences from pupae [29]. Still, other studies on fruit flies have included 0.07–0.3% benzoic acid in the larvae’s feed substrates [30,31] but have not investigated potential effects of this directly. Thus, benzoic acid may potentially be used as a safe antimicrobial additive also in feed substrates for BSF larvae in low concentrations. However, only little seems to be known about how benzoic acid affects BSF larvae, or whether it will affect the microorganisms in the feed substrates at relevant doses. BSF larvae can grow at pH values below the p*K_a_* value of benzoic acid but may not develop as rapidly and not reach as high weights as at higher pH [32,33]. Still, if benzoic acid can slow down microbial substrate degradation without harming the BSF larvae to an unacceptably degree, it may reduce the competition from microorganisms and improve the feed efficiency, and potentially the yield of BSF larvae as well [12]. Therefore, in this study, we investigate the effects of benzoic acid, dosed at low concentrations comparable to those used in the fruit fly studies into the feed substrate of BSF larvae, on the growth and feed efficiency of BSF larvae, and on parallel microbial activities in their feed substrates. The effect of benzoic acid on BSF larvae can be assessed based on NGE*_avg_ as well as the larvae’s specific growth rate and maximum weight, while effects on the microbial activity in the feed substrates will also affect the SCE and SRR. Our aim is to provide an initial assessment of benzoic acid as antimicrobial feed additive for BSF larvae.

## 2. Materials and Methods

### 2.1. BSF Larvae

BSF larvae and eggs were obtained from Enorm Biofactory ApS, Flemming, Denmark. Larvae were used directly as starter larvae upon arrival at our laboratory. Eggs were added into 2.5 kg of chicken feed, DLG Paco Start 19, Danish Agro, Karise, Denmark, with 70% moisture content and incubated at 32 °C and 85–90% relative humidity in a dark climate chamber (Binder B-BD-BF incubator, BINDER GmbH, Tuttlingen, Germany) for 5 days before use as starter larvae.

### 2.2. Feed Substrates

Chicken feed, DLG Paco Start 19 (17% protein, 2–3% fat, and 57–59% carbohydrates based on dry matter, according to the producer), was used as feed substrate (additional information on its composition is found in [7]). The substrate moisture content was adjusted to 72% by addition of deionized water. In some rearing experiments, pH of the feed substrate was adjusted to pH 3.6 or 4 (<p*K_a_* of benzoic acid) by titration of 0.2 M citric acid into partially moist chicken feed. After each dose of citric acid, the moist chicken feed was manually mixed, 1 g of feed substrate was removed and suspended in 2 mL of distilled water, and pH was recorded. The procedure was repeated until pH was ≤4. Additional water was then added to obtain a substrate moisture content of 72%. In some experiments, sodium benzoate (Item number 26060003, Urtegaarden, Frederiksberg, Denmark) was dissolved in the deionized water before it was added to the feed substrates to reach final concentrations of 0.5, 1, or 2 g L^−1^ in the water phase of the substrate, referred to as 0.05%, 0.1%, or 0.2% benzoic acid, the actual antimicrobial compound. Single batches of feed substrate were prepared, divided, and distributed into replicate rearing trials to minimize the variability of feed substrate composition.

### 2.3. Rearing Experiments

BSF larvae were reared in three series of experiments at different pH and concentrations of benzoic acid, as well as at different larval densities. The BSF larvae were reared in cylindrical containers (diameter = 7.5 cm, height = 11.5 cm) placed in a dark climate chamber at 28 °C. The containers were covered by coarse-meshed cloth, secured by rubber bands to pre-vent larvae from escaping. An open water reservoir inside the climate chamber maintained 85–90% humidity.

One series of rearing experiments was carried out at initial substrate pH 3.6 with 50 starter larvae added to each container along with 50 g of wet chicken feed (14.2 g of dried substrate), corresponding to 284 mg of dry chicken feed per larva, and 0–0.2% benzoic acid. Thus, we increased the feed supply above recent recommendations of 220 mg dry feed per BSF larva [34]. Other experiments suggest that close to 300 dry chicken feed is needed to avoid BSF larvae from being feed limited [7]. Rearing trials at pH 7.6 and no benzoic acid were included as control. The starter larvae, 2 days of age weighing 1.1–1.2 mg DW were reared from eggs.

The second series of rearing experiments was carried out at initial substrate pH 7.6 with 50 starter larvae added to each container along with 41.8 g of dry chicken feed, corresponding to 836 mg of dry chicken feed per larva, and 0–0.2% benzoic acid. The starter larvae, 9 days of age weighing 8.2–8.5 mg DW were hatched and initially reared at Enorm Biofactory.

The third series of rearing experiments was carried out at initial substrate pH 4 using four different larval densities of 15, 30, 60, and 120 larvae per 8.4 g of dry chicken feed, corresponding to 557, 278, 139, or 70 mg of dry chicken feed per larva, respectively, with either 0 or 0.2% benzoic acid. Thereby, this series of growth trials spanned feed rations from considerably below to considerably above the recent recommendations of 220–300 mg dry feed per BSF larva [7,34] allowing us to investigate the effect of benzoic acid in feed deficient as well as in feed sufficient BSF larvae. The starter larvae, 5 days of age, weighing 4.6–5.8 mg DW, were hatched and initially reared at Enorm Biofactory. All trials within each series of rearing experiments were carried out in triplicate and conducted in parallel in the same incubator to ensure equal conditions. The experimental set-ups are illustrated in Appendix A, and additional information on the experimental conditions is presented in Appendix A.

### 2.4. CO_2_ Production Rates

CO_2_ production rates from 5 randomly selected BSF larvae as well as from 1 g of wet substrate from each container were measured at regular intervals. Larvae or feed substrate were collected and immediately transferred to a 50 mL plastic centrifugation tube placed in a water bath at 28 °C. A PS2110 Carbon Dioxide Gas Sensor (Pasco, Roseville, CA, USA) was inserted into the top of the tube, thereby forming a closed respiration chamber. The CO_2_ concentration in the respiration chamber was recorded at a frequency of 1 Hz. Linear increases in CO_2_ concentrations recorded for a period of 2 min, starting 1 min after the respiration chamber had been closed, were used as measures of larval CO_2_ production rates. The larvae were then weighed and returned alive to the same container from where they had been collected. The CO_2_ sensor had been calibrated in a 1.18 L closed, cylindrical Perspex chamber with a magnetic stirrer bar at the bottom. The calibration chamber was filled with N_2_. Pulses of 0.2 mL CO_2_ were injected stepwise from a syringe after the signal on the sensor after the previous injection had stabilized. The relationship between the sensor signal and CO_2_ concentration in the measuring chamber is linear from 0 to at least 10,000 ppm. When the CO_2_ production rates from larvae or feed substates were measured, the CO_2_ concentrations reached values up to 3000–4000 ppm, thus remaining within the linear range of the sensor.

### 2.5. Analytical Procedures

The weight of the feed substrate was measured when the rearing experiments were started, and the leftover mixtures of feed substrate and frass were measured when the experiments were terminated. The weights of feed substrate samples BSF larvae were also measured immediately after the measurements of CO_2_ production rates and when the experiments were terminated. Samples of larvae, feed substrates, and frass were dried at 105 °C to quantify moisture contents and weights in terms of dry matter (DM).

### 2.6. Growth Model

Larval growth was modelled as described in detail in [6]. In brief, the Verhulst logistic model was as follows:(1)$$X=\frac{X_{max}}{1+\left(\frac{X_{max}-X_{0}}{X_{0}}\right)e^{-\mu_{max}\left(t-t_{0}\right)}}$$
where *X_max_* is the maximal weight of the larvae (mg), *X*_0_ is the weight of the starter larvae (mg) at *t* = *t*_0_ (onset of rearing experiment), and *μ_max_* is the maximal specific growth rate of the larvae (day^−1^), fitted to the measured DW of the BSF larvae to find the combination of *X_max_* (measured as DW) and *μ_max_* that resulted in the smallest difference (absolute mean error) between model and measured values. The Verhulst logistic model accurately describes growth curves of BSF larvae [6]. Therefore, the growth rate of the larvae, *r_X_* (mg day^−1^) can be estimated from the first order derivative of Equation (1):(2)$$r_{X}=\frac{dX}{dt}=\mu_{max}X\left(1-\frac{X}{X_{max}}\right)$$
from where the specific growth rate, *µ* (day^−1^), was found:(3)$$\mu =\frac{r_{X}}{X}=\mu_{max}\left(1-\frac{X}{X_{max}}\right)$$

The specific growth rates, which were actually attained by the BSF larvae, were estimated from ln-transformed larval weights during the initial growth phase where the increase in larval DW was close to exponential.

Larval CO_2_ production stems from growth as well as maintenance, and the rate of larval CO_2_ production, *r_CO_*_2_ (mg CO_2_ day^−1^) was estimated from Equation (4):(4)$$r_{CO2}=Y\mu X+mX$$
where *Y* (unitless) is the cost of growth, and *m* is a maintenance coefficient (day^−1^). *Y* and *m* are considered constants, and Equation (4) was fitted to measures larval CO_2_ production rates. Feed assimilation rates, *r_A_* (mg day^−1^) were finally estimated as(5)$$r_{A}=r_{X}+r_{CO2}$$

All variables were expressed in terms of carbon equivalents when the model in Equations (1)–(5) was executed, and afterwards converted into units of dry matter, based on the mass fraction of carbon in the larvae, in the feed, and in CO_2_.

### 2.7. Performance Indicators

The carbon net growth efficiency (NGE*) was calculated from Equation (6) [14]:(6)$${NGE}^{*}=\frac{r_{X}}{r_{A}}$$
and the average carbon net growth efficiencies of the larvae across the experiments (NGE*_avg_) was estimated from Equation (7),(7)$${NGE}_{avg}^{*}=\frac{\int r_{X}}{\int r_{A}}=\frac{X_{t}-X_{0}}{\int r_{A}}$$
where *X*_0_ and *X_t_* are larval DW at time zero and at harvest, respectively.

Substrate conversion efficiencies (SCE) were evaluated from a mass balance approach, comparing the overall gain of larval weight to the overall decrease feed, in terms of DM:(8)$$SCE=\frac{n_{t}X_{t}-n_{0}X_{0}}{W_{0}-W_{t}}$$
where *n*_0_ and *n_t_* are the number of larvae at start and at time of harvest, and *W*_0_ and *W_t_* are the total dry weights of the substrates and the substrate/frass mixtures at start and at harvest, respectively. The substrate reduction rate (SRR), was calculated from Equation (9):(9)$$SRR=\frac{W_{0}-W_{t}}{W_{0}}$$

### 2.8. Conversion Factors and Statistical Analysis

In order to compare net growth efficiencies to SCE, NGE*_avg_ was expressed also in terms of dry matter, NGE*_avg,DW_. For this, carbon was assumed to make up 55% of the DM fraction of the larvae [5,7]; 49% of the DM fraction of the organic part of the feed, corresponding to the carbon content in microbial biomass and plant-based materials [35,36]; and 27% of CO_2_.

Maximal larval weight, specific growth rate, total larval weight, NGE*_avg_, SCE, and SRR were compared in terms of a null hypothesis of equal larval performance at all benzoic acid concentrations. Larval performances were considered significantly different when the null hypothesis was rejected by one-way ANOVA analysis at 5% probability level across all benzoic acid concentrations or larval densities, and regression analysis indicated systematic, dose-dependent effects on larval performances. Larval performances at high and low pH levels or zero and 0.2% benzoic acid were compared by *t*-tests at 5% probability level. Previous studies have verified that individual larval dry weights follow Gaussian distributions across their growth phase under various environmental conditions [11,37]. The same was assumed for the other variables.

## 3. Results and Discussion

### 3.1. Effects of Benzoic Acid on Growth and Feed Efficiency at pH 4

Figure 1 shows DW, CO_2_ production rates, and NGE* of BSF larvae grown on chicken feed supplemented by 0–0.2% benzoic acid with initial pH 3.6. Larvae were also reared at 0% benzoic acid at neutral pH, considered control conditions. Growth was, in all cases, well described by logistic curves until the larvae reached their maximal weight 11–13 days of age at neutral pH and 13–15 days of age at low pH. Thereafter, they lost weight. Larval CO_2_ production rates increased with larval weight but decreased again in the last days of the experiments after larval weights had peaked, and the larvae were no longer expending energy on growth [6]. NGE* was highest in the smallest larvae, close to 0.8, and decreased gradually to reach zero when the larvae reached their maximal weight. Key results from the experiments in Figure 1 are listed in Appendix A.

Low concentrations of benzoic acid, as well as low pH, affected the BSF larvae. The strongest effects were caused by the low pH values, although the survival rate stayed above 96% at all conditions (Appendix A). The largest larvae (106 mg DW) were observed at neutral pH (Figure 2A). At pH 4, the larvae reached only 89–102 mg DW. Also, the highest specific growth rate (0.98 day^−1^) was observed at neutral pH (Figure 2B). Lower values were seen at pH 4 (0.74–0.81 day^−1^). Similar differences of the maximal specific growth rates were predicted by Equation (1) (Appendix A). Low larval weight at low pH is consistent with observations from previous studies [32,33] and is probably resulting from a comparatively high metabolic burden. Larval CO_2_ production rates peaked at 30–40 mg day^−1^ at low pH (Figure 1B–E) but only 20–25 mg day^−1^ at neutral pH (Figure 1A). Still, larval DWs close to 100 mg DW remain at the high end of what has been observed in other studies [38]. Thus, the low pH value was compatible with the environmental requirements of the BSF larvae. ANOVA analysis indicated that the maximal larval weights were different at the different benzoic acid concentrations (*p* < 0.01). However, these differences did not depend on the concentration of benzoic acid, and there was no correlation between these variables (*r*^2^ = 0.09). Thus, low concentrations of benzoic acid do not seem to affect the maximal weight of BSF larvae. Benzoic acid also did not affect the total larval DM harvested on day 19 (Figure 2C). Although the specific growth rates seemed significantly different at the different benzoic acid concentrations (*p* = 0.02), this variable was only weakly correlated (*r*^2^ = 0.47) with the benzoic acid concentration. Thus, low concentrations of benzoic acid seem to have no or only minor effects on the specific growth rate of BSF larvae. Finally, was NGE*_avg_ reduced from 0.55 at neutral pH to 0.43–0.50 at pH 4 (Figure 2D). In previous studies on BSF larvae reared on chicken feed, NGE*_avg_ = 0.50–0.57 [6,7,11,37]. This difference also suggests that lowering the pH value in the feeding substrate is not without some metabolic cost for the BSF larvae. NGE*_avg_ did, however, correlate positively (*r*^2^ = 0.70) with the benzoic acid concentration (*p* < 0.01). The increase in NGE*_avg_ with benzoic acid concentration was due to the concomitant lowering of larval CO_2_ production rates, as can be seen in Figure 1. It is not clear why benzoic acid decreased larval CO_2_ production rates. However, it has been reported that low doses of benzoic acid in humans and other mammals can improve gut functions, increase nutrient digestibility, and benefit the gut microbiome [39]. Similarly, low doses of benzoic acid seem to either benefit also the BSF larvae or maybe make more of the most nutritional substrate components available to the larvae.

Specific CO_2_ evolution rates from samples of feed substrates free from BSF larvae are shown in Figure 3A. Pronounced day-to-day variations occurred but rates were generally lowest at the highest concentrations of benzoic acid across the 17-day experimental period. This effect of benzoic acid is seen more clearly in Figure 3B, showing the average specific CO_2_ evolution rates from feed substrates across the experiments. While pH differences did not result in significant differences in the absence of benzoic acid (*p* = 0.51), benzoic acid decreased the specific CO_2_ evolution rate dose dependently (*p* < 0.01, *r*^2^ = 0.90). It should be noted that the measured CO_2_ evolution rates from the small samples of feed substrate removed from the larval cultures may not quantitatively represent the actual microbial activity that were in the larval cultures from where the samples were taken. The measured CO_2_ evolution rates rather indicate the capacity of aerobic microbial metabolism since the surface to volume ratio, and thus the accessibility of oxygen, has been higher in the samples than in the larval cultures. Still, benzoic acid undoubtedly affected the microbial activities in the feed substrates, lowering the capacity to degrade feed substrate components into CO_2_ across the experimental period.

The SRR was unaffected by the pH, reaching 0.76 at neutral pH as well as low pH with no benzoic acid added (*p* = 0.81, Figure 4A). In contrast, benzoic acid resulted in modest but significant differences in the SRR (*p* < 0.01), and the SRR was negatively correlated to the benzoic acid concentration (*r*^2^ = 0.79). Likely, this reflects that not only the microbial capacity (Figure 3), but also the actual microbial activity in the feed substrates, were reduced by benzoic acid. Therefore, less feed substrate was degraded microbially in the presence of benzoic acid. The SCE (Figure 4B) was highest at low pH (*p* = 0.01) and positively correlated with the benzoic acid concentration (*p* < 0.01, *r*^2^ > 0.99). This is because the lowering of the SRR by benzoic acid hardly affected the final weight of the BSF larvae (Appendix A). A higher proportion of the feed substrates were thus degraded by the BSF larvae and used for growth in the presence of benzoic acid. This further indicates that microbial activities had indeed been lowest in the cultures having the highest concentrations of benzoic acid. As expected, the SCE was lower than NGE*_avg_ (Figure 2C), partly because the SCE is affected by microbial substrate degradation and mortality [7,14], and also because NGE*_avg_ was estimated only until the time the larvae reached maximal weight, while SCE was estimated from a mass balance at the end of the experiment when the larvae had lost some weight (Figure 1).

### 3.2. Effects of Benzoic Acid on Growth and Feed Efficiency at Neutral pH

We also conducted a series of experiments, like the ones described above, but without adjusting the pH. The results are shown in Appendix A. The starter larvae used in these experiments had an average DW of close to 8 mg, six times larger than the starter larvae used in the experiment shown in Figure 1, and they were offered a higher feed ration. Therefore, the magnitude of variables and parameters from these experiments cannot be directly compared to those obtained from Figure 1, Figure 2, Figure 3 and Figure 4. Still, larval growth curves followed the logistic model, and their CO_2_ production rates increased with weight. The highest individual CO_2_ production rate of 20 mg day^−1^ was again measured in larvae not exposed to benzoic acid, as compared to 13–18 mg day^−1^ in those exposed to benzoic acid. However, no significant effects of benzoic acid concentration on maximal larval weight, specific growth rate, NGE*_avg_, SRR, or SCE were observed. Specific CO_2_ production rates from the feed substrate samples also varied day by day, but the average specific CO_2_ production rates across the experiments were not significantly affected by benzoic acid. At neutral pH, low concentrations of benzoic acid in the feed substrates (0.05–0.2%) thus appear to be harmless to BSF larvae. As expected from [18], also the microbial CO_2_ production in the feed substrates seemed unaffected by benzoic acid at neutral pH.

### 3.3. Effects of Larval Density

In a third series of rearing experiments, BSF larvae were reared at initial pH 4 with or without 0.2% benzoic acid, the concentration that in the previous experiments had the strongest effect on SCR and SRR (Figure 4) without harming the BSF larvae (Figure 2). The larval densities of 15, 30, 60, and 120 individuals per 8.4 g of dry chicken feed corresponded to 557, 278, 139, or 70 mg of dry chicken feed per larva, respectively. The purpose was to investigate whether reduced microbial activity would preserve more of the food for the BSF larvae, thereby increasing their food availability and increasing their individual weight, in the event they became food-limited before reaching the prepupal stage [15]. Growth curves are shown in Appendix A. Regardless of benzoic acid in the feed substrates, the maximal DW of the larvae was negatively correlated to their density (*p* < 0.01, *r*^2^ = 0.94, Appendix A) and thus positively correlated to the amount of feed available per larva (Figure 5). Similar patterns have also been observed in former studies on BSF larvae reared at different larval densities [2,3,4]. Only the larvae fed 557 mg of chicken feed per larva reached individual DWs around 100 mg, close to the maximal weights that have also been found in other studies [38] and in the experiments shown in Figure 1. This suggests a minimal requirement of around 300 mg chicken feed per larva to avoid feed restriction. More feed is probably needed if less nutritious feed substrates are used. Conversely, the total dry weight of BSF larvae produced was positively correlated to their density (*p* < 0.01, *r*^2^ = 0.93, Appendix A) and negatively related to their individual feed ration (Figure 5B). Again, similar patterns have been seen in previous studies of BSF larvae reared at different densities [2,3,4]. Benzoic acid did not seem to affect the maximal DW of the individual larvae or the total weight of larval DM (Figure 5A,B).

More feed substrate was removed at high compared to low larval densities, and the SRR was highest at high larval densities (Appendix A). Low individual feed rations and feed limitation thus promoted high overall turnover of the feed substrate (Figure 5C). Furthermore, the SRR was lowest in the presence of benzoic acid at all larval densities. At 557 and 278 mg chicken feed per larva, the differences were highly significant (*p* < 0.01), suggesting that these larvae developed into prepupae before all nutritional components of the feed substrate was utilized, and that benzoic acid reduced microbial degradation of the surplus feed substrate. At 139 and 70 mg chicken feed per larva, the BSF larvae became feed restricted before reaching the prepupal state, and the differences in SRR due to benzoic acid were much smaller and barely significant (*p* = 0.04–0.05). Benzoic acid significantly increased SCE at all larval densities (*p* < 0.05). This is also consistent with the fact that relatively less feed substrate was degraded microbially when benzoic acid was present. Since benzoic acid had the strongest effect on SRR at high individual feed rations, the largest differences in SCE due to benzoic acid were also observed here (Figure 5D and Appendix A). High SCE was seen at the lowest feed ration, probably reflecting that the feed-restricted BSF larvae produced most of their biomass, while they were still small and their NGE* was high. SCE was also high at the highest feed ration when benzoic acid was present in the feed substrate. These feed sufficient BSF larvae have had access to the most nutritious feed compounds throughout their lifetime, partly due to the abundance of feed, but also due to reduced competition from microorganisms in the substrate.

## 4. Conclusions

Low concentrations (0.05–0.2%) of benzoic acid had no or only minor effects on the BSF larvae but inhibited microbial activities in the feed substrates at low pH. Less feed substrate was degraded when benzoic acid was present, resulting in decreased SRR. This did not affect how much larval biomass was produced. NGE*_avg_ slightly increased, indicating that the larvae utilized the feed substrate most effectively in presences of benzoic acid. The decrease in SRR and the increase in NGE*_avg_ meant increased SCE when benzoic acid was added. Thus, benzoic acid seems to be harmless to BSF larvae at concentrations up to at least 0.2% in the water phase of the feed substrate, while the microbial activities were suppressed, demonstrating that microbial activities in feed substrates can be targeted selectively by this commonly used feed additive. Yet, the effect of benzoic acid depends on pH values below what is optimal for BSF larvae, and its use may come at a cost. Nevertheless, this study demonstrates that suppression of microbial activities in feed substrates may favor BSF larvae in their competition for nutritional components, potentially benefiting the recycling of organic residues.

## Figures and Tables

**Figure 1 insects-16-01155-f001:**
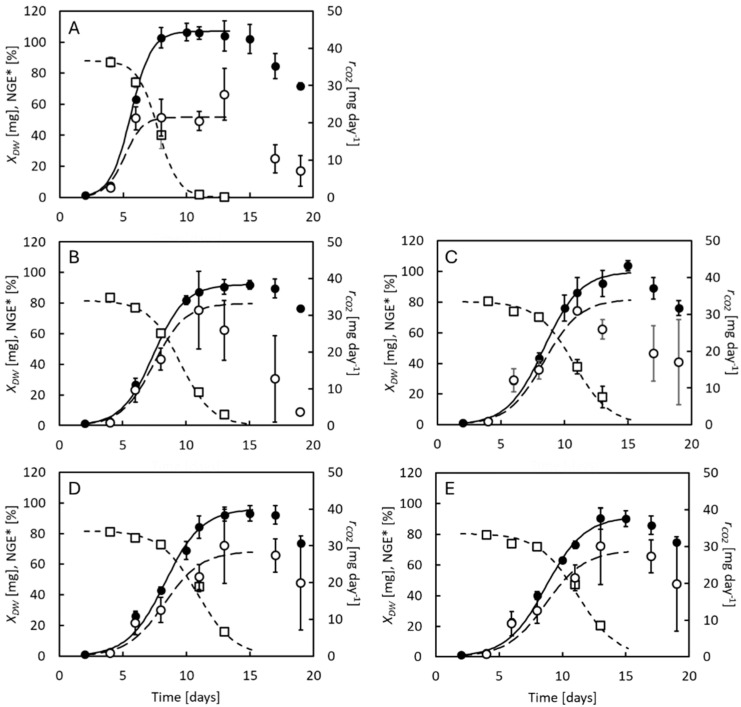
Dry weight (●), CO_2_ production rate (○), and net growth efficiency (□) of BSF larvae reared at different concentrations of benzoic acid. (**A**) Control, 0% benzoic acid and neutral pH. (**B**–**E**) Larvae reared at 0, 0.05%, 0.1%, or 0.2% benzoic acid, respectively, and initial pH 4. Data points indicate average values ± standard deviation of 3 replicate cultures. Curves are modelled by Equations (1), (4) and (6).

**Figure 2 insects-16-01155-f002:**
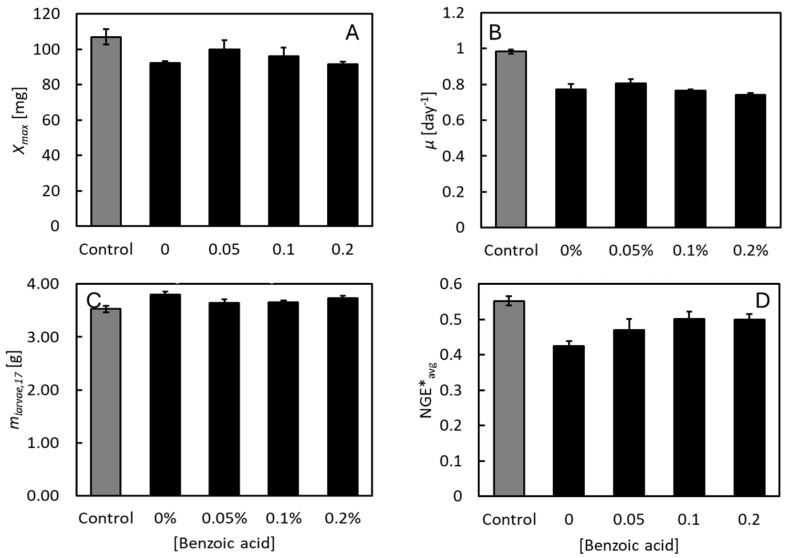
BSF larvae reared at and different concentrations of benzoic acid and neutral pH (grey bars) or low pH (black bars). (**A**) Maximal DW, *X_max_*. (**B**) Specific growth rate during exponential growth phase, *µ*. (**C**) Total harvested larval dry matter on day 19. (**D**) Average carbon net growth efficiency, NGE*_avg_. Bars indicate average values ± standard deviation of 3 replicate cultures.

**Figure 3 insects-16-01155-f003:**
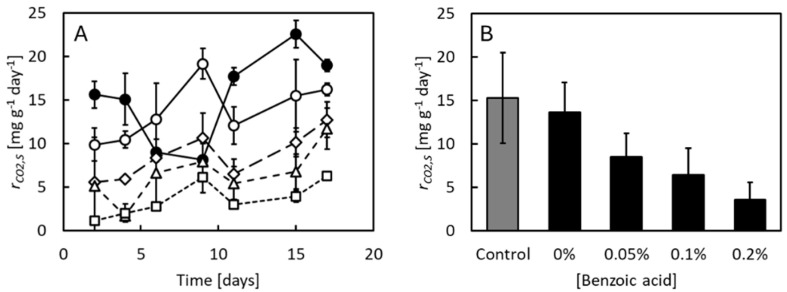
(**A**) CO_2_ production rates from feed substrates, *r_CO*2**,*S_* sampled from cultures of BSF larvae reared at neutral pH and no benzoic acid (●), low pH and no benzoic acid (○), low pH and 0.05% benzoic acid (◊), low pH and 0.1% benzoic acid (Δ), and low pH and 0.2% benzoic acid (□). Data points indicate average values ± standard deviation of 3 replicate cultures. (**B**) Average CO_2_ production rates from feed substrates at neutral pH and no benzoic acid (grey bar) and low pH and different concentrations of benzoic acid (black bars). Bars indicate average values ± standard deviation of 3 replicate cultures.

**Figure 4 insects-16-01155-f004:**
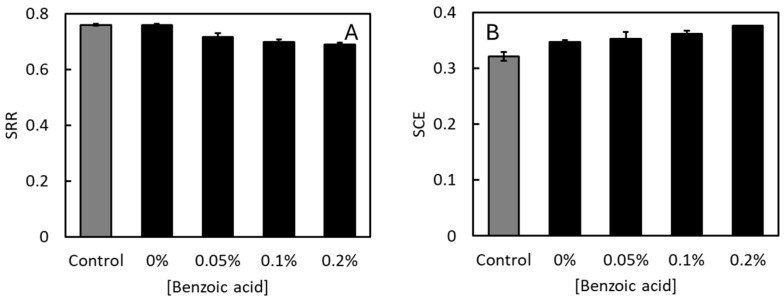
BSF larvae reared at neutral pH (grey bars) and low pH (black bars) and different concentrations of benzoic acid. (**A**) Substrate reduction rate (SSR). (**B**) Substrate conversion efficiency (SCE). Bars indicate average values ± standard deviation of 3 replicate cultures.

**Figure 5 insects-16-01155-f005:**
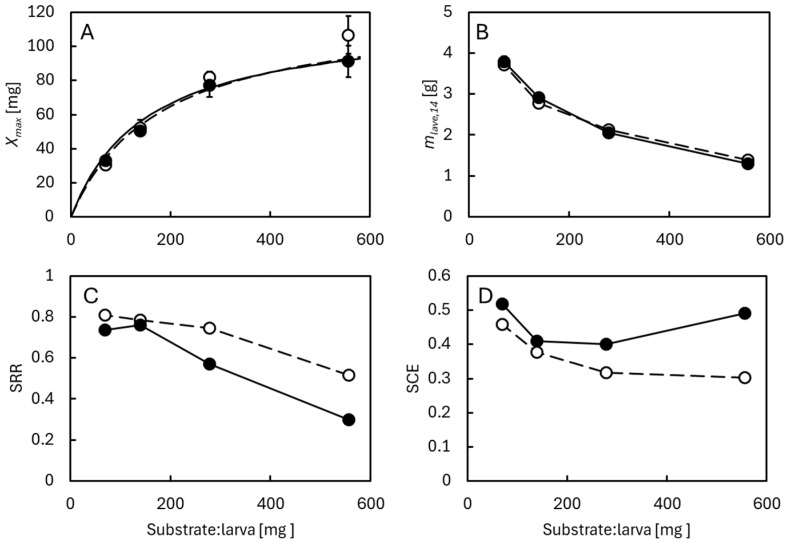
BSF larvae reared at different ratios of chicken feed per larva at initial pH = 4 with no (open symbols, dashed line) or 0.2% benzoic acid (closed symbols, solid line) in the feed substrate. (**A**) Maximal DW, *X_max_*. Curves show best fits of hyperbolic saturation curves [40], indicating that highest *X_max_* ≈ 117–122 mg DW and that 151–177 mg chicken feed must be supplied for BSF larvae to obtain half this DW. (**B**) Total larval dry matter at end of experiment, day 14. (**C**) Substrate reduction rate, SSR. (**D**) Substrate conversion efficiency, SCE. Bars indicate average values ± standard deviation of 3 replicate cultures. Growth curves are shown in Appendix A, and the effects of benzoic acid at each larval density compared in Appendix A.

## Data Availability

The original data presented in the study are openly available in Mendeley Data at https://doi.org/10.17632/872yj62fjr.

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
