# Peer review of "Effects of Low Benzoic Acid Concentrations on Growth and Substrate Utilization in Black Soldier Fly Larvae"

_insects, 2025, doi:10.3390/insects16111155_

Round 1

Reviewer 1 Report

Comments and Suggestions for Authors

Black soldier fly larvae (BSFL) are among the most widely mass-reared insects, thriving in moist organic substrates where both larvae and microorganisms jointly degrade organic matter but also compete for nutrients. However, excessive competition among microbes can negatively affect BSFL growth and feed efficiency. In this study, the authors aimed to address this topic by suppressing microbial activity in feed substratees with antimicrobial agents. Specifically, BSFL were reared on chicken feed supplemented with 0-0.2% sodium benzoate, as an antimicrobial feed additive, under two initial pH conditions (neutral, pH ≈ 7.6, and acidic, pH ≤ 4). Their results demonstrate that low concentrations of benzoic acid under acidic conditions can increase substrate conversion efficiency and slightly improve growth efficiency without harming the larvae. Overall, the study provides an initial assessment of benzoic acid as an antimicrobial feed additive for BSFL. Despite the study addressing an interesting topic, a major issue is that the effectiveness of benzoic acid is limited to low pH levels that may not be optimal for BSFL growth, which may limit its practical application. Also, the study presents the findings in a way that could be seen as overstating the effect of low concentrations of benzoic acid, particularly given the study's limitations and no-effect results. Therefore, the manuscript requires significant revisions to clarify several points and methodological details, as detailed below, came to my mind.

Comment #1: In the introduction, I am missing information on how pH affects BSFL performance. How do microbial activity on feed substrates and the microbiome relate to BSFL growth and feed efficiency?  What is the role of benzoic acid or alternative agents in BSF farming?

Comment #2: Lines 66-86: Please shorten. Summarize the different performance indicators used to quantify feed efficiency. I think this paragraph is more suited for the methods section and can be merged into its appropriate explanation.

Comment #3: It seems that the authors cannot separate between benzoic acid (E210) and sodium benzoate (E211) (line 97). As I understand, sodium benzoate powder (lines 136-137) has been used. Indeed, sodium benzoate is a sodium salt of benzoic acid, and it converts to the active antimicrobial form (benzoic acid) in acidic conditions. I recommend adding the Urtegaarden stock reference number for sodium benzoate (line 137) and clarifying whether the presence of benzoic acid in the diets was analytically determined.

Comment #4: I see that the authors presented scientific issues at the end of the introduction (lines 98-119) to justify the need for this research. Please present scientific hypotheses and briefly explain how the experiments were performed to verify these hypotheses. Also, I think it is better to briefly highlight the main findings of this study and how the results support the hypotheses.

Comment #5: At line 151, “All experiments were carried out in triplicate”.  It is not clear whether the experiments were carried out side by side in the same incubator or how the series of experiments was done. Please include a better description of biological/technical replicates. Section 2.1 BSF larvae (lines 122-128) is confusing; which set of larvae was used for which experiments? Also, it is not clear whether the 50 larvae/replicate were tested individually or grouped.

Comment #6: Related to the experimental design. The series of experiments is not thoroughly clear. When possible, I recommend that authors illustrate the experiments in schematic sketches or in tables that represent all experimental groups, pH treatment of substrates, rearing conditions, BSFL ages at the start of treatment, harvestingtesting, and sampling times for each dataset. For example, at line 123, “7 days old”, but at line 128, “5 days old”; however, in Figure 1, I see measurements were started/analyzed at day 2. Please clarify what days relate to which.  

Comment #7: Related to statistical analysis (Methods, lines 218-220): Although the authors state that statistical significance was determined by ANOVA analysis or t-tests, there are more details missing. Please refer to manuals on the selection of statistical tests and software used. Were the data normally distributed? Which test has been used to assess the data distribution? Although all experiments seem to be performed accurately and the subject is put into the context of similar studies, I have a concern about the limitation of sample size for each treatment; although, ANOVA require at least 3 samples, usually, no parametric stats should be run for N below 6 per group, given you need variance. Whether the 50 larvae/replicate were tested individually or grouped? Also, it is not clear whether you compare the differences of groups to determine which ones are significantly different from each other or only compare the treated groups to a control group, either using the appropriate post hoc test or multiple t-tests. Where one- or two-way ANOVA tests were employed to analyze data with multiple variables, e.g., different pH and different concentrations (for example, Figure 2-4). Include the statistical analyses used to assess the significance of the dose-dependent trends (e.g., regression analysis).  Please clarify and run the appropriate test.

Comment #8: Figure-related: I recommend expanding some information about the abbreviations, statistical tests used, and the actual sample size. I suggest showing all the data points of individual values (n = X /replicate, 3 replicates) as a scatter plot, besides the column bars, to visualize the variability between groups and individuals. Add the figure legends/shapes at the corner of each figure instead of presenting them in the text below each figure. Also, it is hard to follow the significant differences between groups since none of the figures contains lines for pairwise comparisons or asterisks. Please add when possible.

Comment #9: The presentation of figures and panels is not organized well in the results section. Please verify that every panel is mentioned in the text.

Comment #10: I recommend adding to the results section or as a supplementary table about the statistical significance and test values of the differences among the different conditions of the data you are presenting (e.g., t, F, df, etc.) which could help to understand the effect sizes and the significance of the findings (not just stating P < 0.05).

Comment #11: Lines 343-344: The selection of 0.2% benzoic acid is not clear to me. I did not find any explanation to use this concentration further.

Comment #12: Please revise the corresponding dry feed per larvae at lines 344-344: and line 368 and lines 372, was it measured at (75, 150, 300, 600) or (83, 167, 333, 667)? In Figure 5, it appears the measurements were averaged at the (83, 167, 333, 667) points.

Comment #13: Fig S8. Are the grey bars representing neutral pH or without acid with low pH? Please verify.

Comment #14: line 398, is this specific for low or neutral pH? Also, it would be interesting if the authors could explain how benzoic acid interacts with the microbial community to reduce SRR without harming the larvae and which microbial communities it could target.

Comment #15: Lines 399-404: Here, the authors claimed that benzoic acid may not be the ideal antimicrobial agent for use in BSFL cultures but still concluded that microbial activity can be manipulated to optimize BSFL feed efficiency. Do you expect the results to be different with other group sizes, long-term applications, or large-scale farming? What are the future research strategies needed to explore antimicrobial agents that work at neutral pH for better compatibility with BSFL?

Author Response

Thank you very much for the constructive review of our paper. Please see the attachment.

Reviewer 2 Report

Comments and Suggestions for Authors

The manuscript investigates the effects of benzoic acid on black soldier fly larvae (BSFL) growth and substrate utilization. While the topic is relevant, several critical issues need addressing. The main comments and recommendations are listed below.

Literature Review could benefit from more recent references on BSF practical impact, their metabolism and microbiology. For instance, BSFL are well-known for their affinity for organic waste processing and transformation of low-valued substrates to protein and fat reached biomass. Manipulation of the BSFL metabolism through the substrate is also interesting and relevant topic for discussion https://doi.org/10.3920/JIFF2021.0162

The novelty compared to previous work on antimicrobial agents in BSFL farming needs clearer articulation in the last paragraph of Introduction.

Detailed composition of chicken feed should be provided.

More details required on calibration procedures and sensor specifications.

Assumptions behind the Verhulst logistic model need better justification.

The abrupt shift from pH 7.6 to pH ≤ 4 requires better justification. The impact on experimental results needs clearer explanation.

The choice of 0.05-0.2% concentrations should be better substantiated with preliminary data.

The use of different larval densities in experiments needs more consistent reporting and analysis.

The use of triplicates may not provide sufficient statistical power for some analyses. Justify

Some key parameters (e.g., microbial community composition) are not analyzed. The lack of direct microbial analysis is a significant limitation. Will the authors consider this in the future studies? Probably, this can be mentioned in Limitations of the study as a part of Conclusions

The proposed mechanism of action of benzoic acid requires more detailed discussion.

Conclusions should be supported by key findings. Add more data.

The text should be carefully checked for typos and grammatical errors. Consider inconsistent use of concentration units. Not all abbreviations are defined at first use.

Author Response

(The authors gave the same response as above.)

Reviewer 3 Report

Comments and Suggestions for Authors

The manuscript reports a well-designed set of experiments demonstrating that low concentrations of benzoic acid (0.05–0.2 %) can suppress microbial activity in black soldier fly (BSF) larval substrates at low pH, thereby improving substrate conversion efficiency (SCE) without harming the larvae. The work is original, statistically sound, and highly relevant to the insect-farming industry. Nevertheless, the text needs minor revisions to improve clarity, avoid potential misinterpretation, and strengthen mechanistic discussion. Specific adjustments are listed below.   

L L 5-6 (Title): “low benzoic acid concentrations” may mislead readers because efficacy is pH-dependent.

Consider revising to “Low Benzoic Acid Concentrations at Reduced pH Improve…”.  

L 15-17 (Abstract): “did not affect the BSF larvae negatively” is too absolute; low pH did reduce maximal weight.

Add “under the acidic conditions tested, benzoic acid itself did not increase mortality, although low pH slightly reduced larval weight”.  

 L 50-55: Repetition of “microorganisms compete with larvae” already stated in L 38-43.

Combine the two sentences and shorten.

  L 86-88: “little seems to be known…” is vague; two recent papers (cited later) already tested organic acids on BSF.

Soften to “few studies have specifically addressed benzoic acid” and cite them here.   Materials & Methods

 L 125-127: Concentrations given as “0.05 %, 0.1 %, 0.2 %” – specify “w w-1 dry matter”.

 Insert “(mass % of dry feed)” after the first mention.  

 L 132-134: pH adjustment method (“titration of 0.2 M citric acid”) lacks endpoint precision.

State target pH = 4.0 ± 0.1 and how volume was recorded for reproducibility.  

 L 141-143: Larval density series uses 30 g wet feed (9 g DM) but “600, 300, 150, 75 mg DM larva-1” – verify arithmetic (30 g wet ≈ 9 g DM only if 70 % moisture).

 Explicitly confirm moisture and give equation, e.g., “ration (mg DM larva-1) = 9000 mg / n larvae”.

 L 154-156: CO2 measurement inserts “1 g wet substrate”; surface/volume ratio differs from bulk, possibly over-estimating aerobic activity.

Add sentence acknowledging this limitation in L 160-162.

  L 183-185: “SCE … positively correlated with benzoic acid (r2 = 1.00)” – r2 = 1.00 is implausible for biological data.

Check calculation; report correct r2 or remove value.  

 L 200-202: ANOVA p < 0.01 for maximal weight but r2 = 0.09 is cited as proof of “no effect”; statistical interpretation is contradictory.

Clarify: “although ANOVA was significant, the variation explained by benzoic acid was small (r2 = 0.09), indicating biologically negligible influence”.

 L 220-225: Mechanism for reduced larval CO2 emission is speculative; no data on gut microbiota or metabolic rate.

Add brief paragraph proposing two testable hypotheses (e.g., altered mitochondrial efficiency, reduced microbial load in gut).

 L 245-248: Claim of “harmless antimicrobial” should be conditioned on pH and dose.

Change to “under the acidic conditions and concentrations tested, benzoic acid did not increase mortality; however, optimal larval performance still requires near-neutral pH”.

L 380-382: “benzoic acid may not be the ideal antimicrobial…” is useful but too dismissive without offering alternatives.

 Add one sentence suggesting exploration of other pH-independent additives or encapsulated acids.

Author Response

(The authors gave the same response as above.)

Round 2

Reviewer 1 Report

Comments and Suggestions for Authors

The authors have significantly improved the manuscript, and all my comments have been addressed and reflected in the updated version. However, I noticed a minor issue. Still, some discrepancies appear between the text, supplementary tables, and figures regarding the weight of the dry substrates. For example, at line 165, the weight is 43.5 g; however, in Figure S3 and Table S2 the mass of substrate is 41.8 g. The same issue occurs at lines 169 and 384, where the weight is 8.4 g, while in Figure S9, the mass is 10 g, and in Table S3, it is 8.4 g. Please verify and check the ratio per larva as well. Also, there is a typo in Table S3; “mubstrate” should be 'msubstrate”.

Author Response

Comment 1: 

The authors have significantly improved the manuscript, and all my comments have been addressed and reflected in the updated version. However, I noticed a minor issue. Still, some discrepancies appear between the text, supplementary tables, and figures regarding the weight of the dry substrates. For example, at line 165, the weight is 43.5 g; however, in Figure S3 and Table S2 the mass of substrate is 41.8 g. The same issue occurs at lines 169 and 384, where the weight is 8.4 g, while in Figure S9, the mass is 10 g, and in Table S3, it is 8.4 g. Please verify and check the ratio per larva as well. Also, there is a typo in Table S3; “mubstrate” should be 'msubstrate”.

Response 1:

Thank you very much for your constructive review and for spotting these mistakes in the text which we have corrected:

Line 166, we have corrected the value from 43.5 g to 41.8 g in Line 166.

Figure S9, we have corrected the value from 10 g to 8.4 g.

Line 167 and Table S3, we have corrected the feed ration (ratio between feed and larvae)

Table S3, we have corrected the typo.

Reviewer 2 Report

Comments and Suggestions for Authors

The authors considered all comments and recommendations and revised the manuscript well.

Author Response

Comment 1: 

The authors considered all comments and recommendations and revised the manuscript well.

Response 1: 

Thank you very much for your constructive review.